# Preperitoneal Pelvic Packing versus Angioembolization for Patients with Hemodynamically Unstable Pelvic Fractures with Pelvic Bleeding: A Single-Centered Retrospective Study

**DOI:** 10.3390/healthcare11121784

**Published:** 2023-06-16

**Authors:** Seong Chan Gong, Ji Eun Park, Sooyeon Kang, Sanghyun An, Myoung Jun Kim, Kwangmin Kim, In Sik Shin

**Affiliations:** 1Department of Surgery, Yonsei University Wonju College of Medicine, Wonju 26426, Republic of Korea; surgeon_g@yonsei.ac.kr (S.C.G.); uldura@yonsei.ac.kr (S.A.); romans_828@yonsei.ac.kr (M.J.K.); 2Department of Medicine, Yonsei University Wonju College of Medicine, Wonju 26426, Republic of Korea; jepark324@yonsei.ac.kr (J.E.P.); k-sy@yonsei.ac.kr (S.K.); 3Graduate School, Yonsei University Wonju College of Medicine, Wonju 26426, Republic of Korea; lukelike@yonsei.ac.kr

**Keywords:** pelvis, fracture, hemorrhage, angiography

## Abstract

The aim of this study was to compare the outcomes of preperitoneal pelvic packing (PPP) and angioembolization (AE) for patients with equivocal vital signs after initial resuscitation. This single-centered retrospective study included information from the database of a regional trauma center from April 2014 to December 2022 for patients with pelvic fractures with a systolic blood pressure of 80–100 mmHg after initial fluid resuscitation. The patients’ characteristics, outcomes, and details of AE after resuscitative endovascular balloon occlusion of the aorta (REBOA) placed in zone III were collected. The follow-up duration was from hospital admission to discharge. A total of 65 patients were enrolled in this study. Their mean age was 59.2 ± 18.1 years, and 40 were males. We divided the enrolled patients into PPP (*n* = 43) and AE (*n* = 22) groups. The median time from emergency department (ED) to procedure and the median duration of ED stay were significantly longer in the AE group than in the PPP group (*p* ≤ 0.001 for both). The median mechanical ventilation (MV) duration was significantly shorter (*p* = 0.046) in the AE group. The number of patients with complications, overall mortality, and mortality due to hemorrhage did not differ between the two groups. Three patients (13.6%) were successfully treated with AE after REBOA. AE may be beneficial for patients with hemodynamically unstable pelvic fractures who show equivocal vital signs after initial fluid resuscitation in terms of reducing the MV duration and incidence of infectious complications.

## 1. Introduction

Pelvic fractures are present in 10% of all patients admitted with blunt trauma; of these, pelvic fracture-related hemodynamic instability is encountered in up to 13% of patients [1,2]. Pelvic fractures with hemodynamic instability represent a significant challenge for trauma surgeons in terms of life threat and functional outcomes. Multidisciplinary approaches to the treatment of hemodynamically unstable pelvic fractures have resulted in improved outcomes for these complex and challenging injuries [3]. However, mortality from pelvic trauma remains as high as 40%, with one-third of patients dying owing to uncontrolled hemorrhage [4]. Various therapeutic modalities, such as temporary pelvic-binding devices, angioembolization (AE), external fixation (EX-FIX) of the pelvis, preperitoneal pelvic packing (PPP), ligation of internal iliac arteries, and resuscitative endovascular balloon occlusion of the aorta (REBOA) have been used in various combinations to stop bleeding [5].

PPP and AE are the main therapeutic strategies for patients with hemodynamically unstable pelvic fractures among the treatment modalities mentioned above. The Eastern Association for the Surgery of Trauma pelvic fracture hemorrhage practice management guidelines state that PPP is an effective technique for controlling pelvic hemorrhage [6]. PPP was first described in Germany in 1994 [7] and is a quick and effective procedure for the emergency control of pelvic hemorrhage that directly addresses bleeding originating from the retroperitoneal space in severe pelvic injuries. However, concerns have been raised regarding the possibility of infectious complications. The guidelines [6] also made level I recommendations to use AE as a bleeding control intervention in hemodynamically unstable pelvic fractures. Pelvic AE is widely used in trauma centers to control arterial sources of intrapelvic bleeding [8]. However, angiographic interventions are not available in all hospitals, and even when available, the angiographic procedure is time-consuming. Therefore, PPP or AE should be chosen by considering the availability of institutional resources, such as proficient trauma surgeons or an angiosuite. In addition, in a well-equipped trauma center where both can be performed, PPP or AE may be chosen based on the patient’s vital signs and conditions. Previous studies have indicated that PPP is the first choice in hemodynamically unstable patients, whereas AE could be the first intervention in stable patients [9,10]. The decision to perform PPP or AE may not be difficult in patients with aggravated hemorrhagic shock despite initial fluid resuscitation and stable vital signs. However, the decision to perform PPP or AE as a first intervention may be difficult for patients showing equivocal vital signs after initial fluid resuscitation. 

A few studies have compared PPP with AE [11,12,13,14]. However, direct comparison of efficacy between PPP and AE may have been insufficient because the number of enrolled patients in some of these studies was small, and baseline characteristics, including vital signs in PPP and AE groups, differed. Furthermore, confounding factors such as AE following PPP existed in some of these studies. We hypothesized that AE may be more beneficial than PPP in patients with pelvic fractures showing equivocal vital signs after initial resuscitation because AE is a relatively less invasive procedure. Therefore, the aim of this study was to directly compare the outcomes of PPP and AE for patients with equivocal vital signs after initial resuscitation. 

## 2. Materials and Methods

### 2.1. Patient Selection and Data Collection

This retrospective study was approved by the institutional review board of Wonju Severance Christian Hospital (IRB no. CR 321179). Medical data of trauma patients were collected from the Korean Trauma Data Bank from 1 April 2014 to 31 December 2022. The data were collected prospectively and analyzed retrospectively. Informed consent was waived because the data were analyzed anonymously. Patients were included in the study if they: (1) visited our trauma center owing to a hemodynamically unstable pelvic fracture (systolic blood pressure [SBP] < 80 mmHg); (2) were aged ≥ 18 years; and (3) agreed to the collection and use of their medical information. Hypotension at initial presentation was defined as SBP < 80 mmHg at the initial emergency department (ED) visit, based on our treatment experience for hemodynamically unstable patients with pelvic fractures. The exclusion criteria were as follows: (1) resuscitation without an initial hemostatic procedure, (2) concomitant PPP and AE, and (3) SBP <80 or >100 mmHg after resuscitation. Based on these criteria, 65 patients were enrolled in this study. The enrollment process for patients with trauma is summarized in Figure 1. 

Variables collected from the database included age, sex, and SBP; worst SBP in ED; SBP after resuscitation (defined as first SBP measured 30 min after the initial visit); pulse rate (PR); cardiac arrest in ED; history of diabetes mellitus; anticoagulant use; injury mechanism; associated injury (abbreviated injury scale [AIS] score ≥ 3); AIS, injury severity score (ISS); initial hemoglobin level; initial lactate level; pelvic fracture type (Young-Burgess classification and Tile classification); and presence of an open fracture. Data on patient management included the use of resuscitative endovascular balloon occlusion of the aorta (REBOA) placed in zone III, a hybrid room, concomitant explo-laparotomy, and open reduction with internal fixation (ORIF). Information collected included the time from injury to ED visit; time from ED visit to intensive care unit (ICU) transfer; time from ED visit to procedure (AE or PPP); time from ED visit to ORIF; duration of ED stay; length of stay (LOS) in hospital; LOS in ICU; packed red blood cell (RBC) transfusion within 24 h after injury; duration of mechanical ventilation (MV); complications, including infectious complications; mortality; and mortality due to hemorrhage. Details of patients who underwent AE after REBOA, including recovered SBP after balloon inflation, time from REBOA insertion to removal, time from REBOA inflation to deflation, location of REBOA ballooning, and embolized artery, were also collected. 

### 2.2. Patient Management

The Wonju Severance Christian Hospital created a trauma team in 2010, and the trauma center was constructed in January 2015. Until January 2014, pelvic binder application, massive transfusion, and pelvic angiography were performed on patients with pelvic fractures with hemodynamic instability who showed contrast extravasation on computed tomography scans. After January 2014, PPP was used for these cases. The treatment protocol for patients with hemodynamically unstable pelvic fractures was established in January 2014 (Figure 2). Resuscitation with 2000 mL crystalloids and no matched blood product was applied to patients with hemodynamically unstable pelvic fractures. The trauma team was activated and engaged in patient management within 10 min. The trauma team consisted of trauma specialists, including emergency medicine physicians, and general, neuro-, orthopedic, and thoracic surgeons. A pelvic binder was applied by the trauma team in the ED to reduce pelvic volume. A trauma series and extended focused assessment with sonography in trauma were performed. REBOA was selectively performed on patients based on the clinical judgement of the trauma physician on duty. If hypotension (SBP < 80 mmHg) in patients was still ongoing after resuscitation, PPP was performed. If SBP was equivocal (80–100 mmHg), computed tomography was considered. In addition, the decision to perform AE or PPP for each patient was determined through a comprehensive assessment that took into account various factors. These included symptoms such as pain and agitation, physical examination data such as tenderness, laboratory data such as hemoglobin and lactate levels, and X-ray findings, including any CT scans conducted to evaluate for signs of extravasation. When the SBP of the patients recovered to above 100 mmHg, CT was performed. When extravasation was observed on CT scans, an AE was performed. Otherwise, patients were managed conservatively. After initial transfusion with two units of O-negative packed RBCs, cross-matched packed RBCs and fresh frozen plasma were administered in a 1:1 ratio as needed, according to our institution’s massive transfusion protocol. The ORIF timing was determined by an orthopedic surgeon in the trauma team. 

### 2.3. PPP Technique

PPP was performed by trauma surgeons who successfully completed the Definitive Surgical Trauma Care training provided by the International Association for Trauma Surgery and Intensive Care. After creating a 7–8 cm vertical skin incision beginning at the symphysis pubis, the anterior sheath of the rectus abdominis muscle was incised, and the muscle was split. After the preperitoneal space was bluntly dissected in the posterolateral direction and the peritoneum was moved to the medial side, the lower border of the sacroiliac joint (SI) was examined. Three surgical tapes were sequentially packed from the lower border of the SI joint using ring forceps. The procedure was repeated on the contralateral side. After correction for coagulopathy, hypothermia, and metabolic acidosis, the packed surgical tapes were removed.

### 2.4. AE

Two interventional radiologists performed angiographic procedures during the study period. If extravasation related to pelvic fracture was observed on contrast CT scans (Figure 3), angiography was decided. If AE was decided for the initial hemostatic procedure, the interventional radiologist on duty was called. Angiography was performed to identify the bleeding artery (Figure 4A). The bleeding artery was selected and embolized to avoid complications related to extensive embolization (Figure 4B). If multifocal bleeding was observed from the internal iliac artery or if the patient’s vital signs worsened without evidence of bleeding on angiography scans, unlike CT findings, nonselective embolization was performed after discussion with a trauma team member. Coils, glue, and gelfoam were used for embolization. A concomitant procedure was planned in the ED, and a hybrid operating room (OR) was used. 

### 2.5. Study Setting

The enrolled patients were divided into PPP and AE groups. Baseline characteristics, factors associated with treatment, time factors, and outcomes were compared. 

### 2.6. Statistical Analysis

Continuous variables are presented as mean ± standard deviation or median (interquartile range) and categorical variables as frequencies and percentages. Continuous variables were tested for normal distribution using the Shapiro–Wilk test and compared using Student’s *t*-test or the Mann–Whitney test, as appropriate. Categorical variables were also tested for normality and compared using the chi-square test and Fisher’s exact test, as appropriate. Statistical significance was set at *p* < 0.05. Statistical analysis was performed using R statistical software (version 4.1.0; R Foundation for Statistical Computing, Vienna, Austria). 

## 3. Results

### 3.1. Patient Enrollment

A total of 65 patients were enrolled in this study. The mean age was 59.2 ± 18.1 years and ranged from 18 to 93 years. The number of males was 40, and 25 were females. The patients were divided into two groups: 22 were classified in the AE group and 43 in the PPP group (Figure 1). Arterial blush was observed in all of the 22 patients who underwent AE. The internal iliac artery branch was embolized in 20 patients. Nonselective embolization of the internal iliac artery was performed in two patients: the ipsilateral internal iliac artery was embolized in one, and the bilateral internal iliac artery was embolized in the other.

### 3.2. Differences in Characteristics of the Enrolled Patients between the Two Groups 

The age (58.3 ± 18.2 vs. 59.6 ± 18.3 years, *p* = 0.782) and proportion of males (13 (59.1%] vs. 27 (62.8%], *p* = 0.983) did not differ between the two groups. Initial SBP (78.8 ± 30.3 vs. 75.3 ± 32.6, *p* = 0.668), worst SBP in ED (66.3 ± 22.2 vs. 63.8 ± 20.9, *p* = 0.663), SBP after fluid resuscitation (90.8 ± 7.7 vs. 92.1 ± 7.0, *p* = 0.512), and PR (105.5 [91.2, 121.5] vs. 110.0 [97.5, 126.0], *p* = 0.637) were not significantly different between the two groups. Associated injury (AIS ≥ 3, 18 (81.8%] vs. 36 (83.7%], *p* = 1.000) and ISS (34.0 [27.5, 38.0] vs. 38.0 [29.0, 44.0], *p* = 0.188) were not different between the two groups. Initial hemoglobin level (11.7 ± 2.4 vs. 10.7 ± 2.5, *p* = 0.108) and pelvic fracture type according to the Young–Burgess classification (*p* = 0.282) and tile classification (*p* = 0.281) did not differ between the two groups. The use of REBOA placed in zone III did not differ between the two groups [3 (13.6%) vs. 2 (4.7%), *p* = 0.326]. External fixation was performed on nine patients in the PPP group and one patient in the AE group, with no statistically significant difference between the groups (*p* = 0.145) (Table 1).

### 3.3. Comparison of Time Factors and Patient Outcomes

Time from ED visit to procedure (139.0 [95.0–155.2] vs. 63.0 [47.0–95.0], *p* < 0.001) and duration of ED stay (124.0 [84.2–221.5] vs. 59.0 [43.5–96.0], *p* < 0.001) were significantly longer in the AE group than in the PPP group. No significant difference was found between the groups for hospital LOS (29.5 [4.8–49.8] vs. 26.0 [8.5–63.0], *p* = 0.901) and ICU LOS (6.0 [2.2–12.8] vs. 9.0 [4.0–16.5], *p* = 0.230). The duration of MV was significantly shorter in the AE group than in the PPP group (2.0 [0.2–8.0] vs. 6.0 [2.5–14.0], *p* = 0.046). The numbers of patients with complications (10 [45.5%] vs. 20 [46.5%], *p* = 1.000), infectious complications (4 (18.2%] vs. 15 (34.9%], *p* = 0.266), mortality (7 (31.8%] vs. 16 (37.2%], *p* = 0.876), and mortality due to hemorrhage (5 (22.7%] vs. 6 (14.0%], *p* = 0.487) did not differ between the two groups (Table 2). 

### 3.4. Details of Patients Who Underwent AE after REBOA 

REBOA was inserted and placed in zone III in three patients with hemodynamically unstable pelvic fractures before AE. The initial SBP was 64 mmHg, and the worst SBP in the ED was 54 mmHg in one patient. After REBOA insertion, AE was successfully performed on the internal iliac branch after SBP recovered. The times from REBOA inflation to deflation in the three patients were 51, 38, and 14 min, respectively, and ISSs were 38, 25, and 22, respectively. The pelvic fracture type was anterior–posterior compression in one patient and lateral compression in the other two. The volumes of RBC transfusions after 24 h were 8, 9, and 2 units, respectively. Lung-associated complications, such as pneumonia and atelectasis, occurred in two patients, and both survived (Table 3).

## 4. Discussion

This study was conducted to compare the outcomes of PPP and AE for patients with equivocal vital signs after initial resuscitation. The time from the ED visit to the procedure was significantly longer in patients who underwent AE. This delay may be attributed to the on-call interventional radiology team having to come to our hospital from outside and the time taken to prepare for angiography at night. The latter could explain the longer ED stays in the AE group. Other studies have also showed similar results [11,12,15]. Hsu, Yadev and Faraj [11] reported that the time to the primary procedure was 67.6 min in PPP and 130.2 min in AE. Li, Dong, Yang, Wang, Wang, Liu, Robinson and Zhou [15] reported a significant difference in time to the primary procedure (77 min in PPP and 102 min in angiography). Therefore, it is reasonable that PPP should be the first line of treatment for patients with persistent hemodynamically unstable pelvic fractures, and AE should be performed as a primary procedure in patients who respond to initial resuscitation, as they may withstand the wait for angiography preparation. 

Our study showed that the outcomes of patients who underwent AE were not inferior to those of patients who underwent PPP. The MV duration was significantly shorter in the AE group than in the PPP group. This was because sedation and MV continued until second-look surgery in patients who underwent PPP. Osborn et al. [16] reported that the MV duration was not significantly different between PPP and AE groups. This discrepancy may have occurred because patients with more severe injuries and lower initial SBP were enrolled in their study; thus, a longer MV was necessary for those who underwent AE than that in our study. Longer MV duration can lead to an increased risk of infections and other complications. Among these, pneumonia is the most common, with MV duration being significantly associated with ventilator-associated pneumonia (VAP) [17,18,19]. An Egyptian study found that the incidence of VAP increased from 5% in patients receiving 1 d of MV to 65% in patients receiving 30 d of MV [19]. Therefore, a shorter MV duration after AE may be beneficial for reducing the risk of pneumonia, although the actual rate of pneumonia development was not significantly different between AE and PPP in this study (4 (18.2%] vs. 11 (25.6%], *p* = 0.720). 

The rate of infectious complications was relatively higher in the PPP group, although the difference was not statistically significant. This may be attributed to surgical site infections (SSI) after PPP. The reported SSI rate after PPP is between 10 and 28% [15,20]. SSI were observed in eight (18.6%) patients in the present study. One of the major infectious complications expected in patients who underwent AE was gluteal necrosis. Previous studies have shown gluteal necrosis after nonselective AE in the internal iliac artery in patients with hemodynamically unstable pelvic fractures [13,21]. Nonselective embolization of the internal iliac artery was performed in four patients in this study. However, gluteal necrosis was not observed. 

The ICU and hospital LOS did not differ between the two groups in this study. Hsu, Yadev and Faraj [11] reported longer ICU and hospital LOS in patients with PPP than in those with AE, whereas other studies showed no difference [15,16]. This discrepancy may have occurred because all these studies were conducted on a relatively small number of patients, the characteristics of the enrolled patients in each study differed, and the hospitals may have had different treatment strategies.

Moreover, mortality and hemorrhagic mortality rates did not differ between the two groups in this study. Hundersmarck, Hietbrink, Leenen and Heng [13] reported a higher mortality rate in patients who underwent PPP, with seven vs. zero exsanguination-related mortalities between the PPP and AE groups. Other studies also revealed no difference in mortality between patients who underwent PPP and AE [11,12,15,16]. However, these results should be interpreted considering the patients’ baseline characteristics. In three of these studies, SBP and ISS at the initial presentation differed between the two groups. Despite this consideration, a difference in mortality rate between the PPP and AE groups was not found in four of the studies, including our study. 

In our study, REBOA placed in zone III was used in three patients with AE and two with PPP. Recently, various studies have indicated the effectiveness of REBOA [22], and some studies have shown REBOA application in patients with pelvic fractures with life-threatening hemorrhage [23,24,25]. Pieper, Thony, Brun, Rodiere, Boussat, Arvieux, Tonetti, Payen and Bouzat [24] reported that SBP increased significantly from 60 to 114 mmHg after balloon inflation in patients with extreme hemodynamic instability with an initial SBP of <60 mmHg. They also reported that the mortality rate was 59%, and complications, such as limb ischemia, iatrogenic aortic dissection, acute kidney injury, and rhabdomyolysis, occurred in patients who underwent REBOA. Although there were only three patients, our study showed that patients were successfully treated with AE after zone III REBOA. After balloon inflation, SBP recovered to 88–100 mmHg, and no mortality was reported in these patients. In addition, treatable complications, such as pneumonia and atelectasis, were reported in two patients. 

In our study, 2000 mL of crystalloids were infused for the initial resuscitation of patients with hemodynamically unstable pelvic fractures. When we first applied this protocol at our trauma center, we implemented this initial resuscitation by referring to the Denver group protocol [26]. This initial resuscitation strategy was previously used in trauma centers in Italy and Hong Kong [12,14]. In our hospital, this initial resuscitation protocol was continuously used even though 2000 mL of crystalloid infusion is no longer the standard treatment. Our protocol may also need modification in line with the current trend. 

This study is meaningful because the impact of the AE was directly compared with that of PPP by excluding patients who underwent both hemostatic procedures (AE and PPP). In addition, we provided evidence on which hemostatic procedure was superior for patients with equivocal vital signs after initial resuscitation by excluding non-responders to initial resuscitation, unlike other studies [13,15,16] that analyzed outcomes in patients with different severities or vital signs. 

This study has several limitations. First, it was based on retrospective analysis of the records of patients treated at a single institution. Second, this study is limited by the small number of patients with hemodynamically unstable pelvic fractures. A small number of patients were enrolled in this study because the prevalence of pelvic fractures with hemodynamic instability was low. However, unidentified heterogeneity in both patient groups due to the limited number of patients may have been present over a longer period, although we attempted to reduce the heterogeneity of patients by selecting those who recovered after initial fluid resuscitation. Third, while we diligently adhered to our treatment algorithm to a great extent, it is crucial to acknowledge that patients received care from multiple physicians throughout the study period. Moreover, certain patients underwent medical interventions such as the administration of tranexamic acid and inotropic agents during the initial presentation. Consequently, these factors have the potential to confound the results and should be taken into consideration. Lastly, as PPP and AE techniques may differ between hospitals or countries, caution is warranted when extrapolating our results to clinical situations with different resources, potentially leading to more or less effective hemorrhage control capabilities and thus affecting outcomes.

## 5. Conclusions

It may be reasonable to conclude that AE can be performed as a primary procedure on patients who respond to initial resuscitation, as they may withstand the preparation time for angiography. The time to the initial hemostatic procedure was significantly longer in AE. AE may be beneficial for patients with hemodynamically unstable pelvic fractures who show equivocal vital signs after initial resuscitation, reducing MV duration and the incidence of infectious complications. In addition, AE after REBOA as a bridge treatment may be a good therapeutic strategy for patients with pelvic fractures with hemodynamic instability. Therefore, further studies with larger numbers of patients included from multiple centers are necessary to confirm our study results. 

## Figures and Tables

**Figure 1 healthcare-11-01784-f001:**
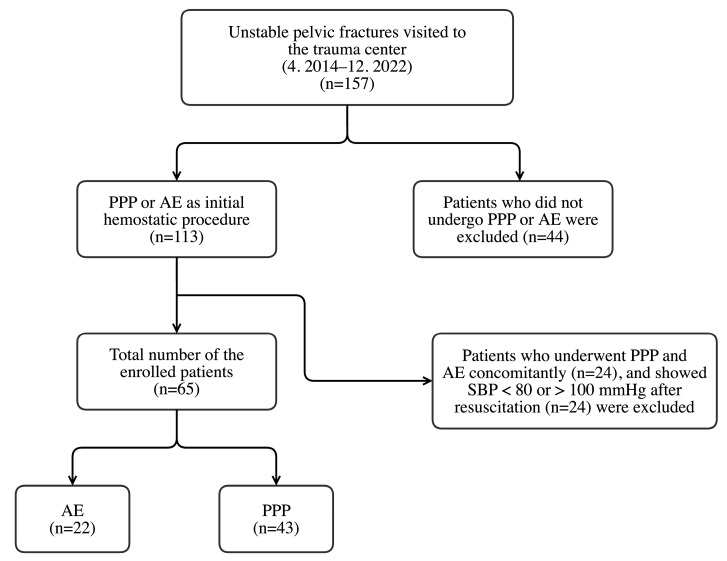
Patient flow chart. PPP, preperitoneal pelvic packing; AE, angioembolization; SBP, systolic blood pressure.

**Figure 2 healthcare-11-01784-f002:**
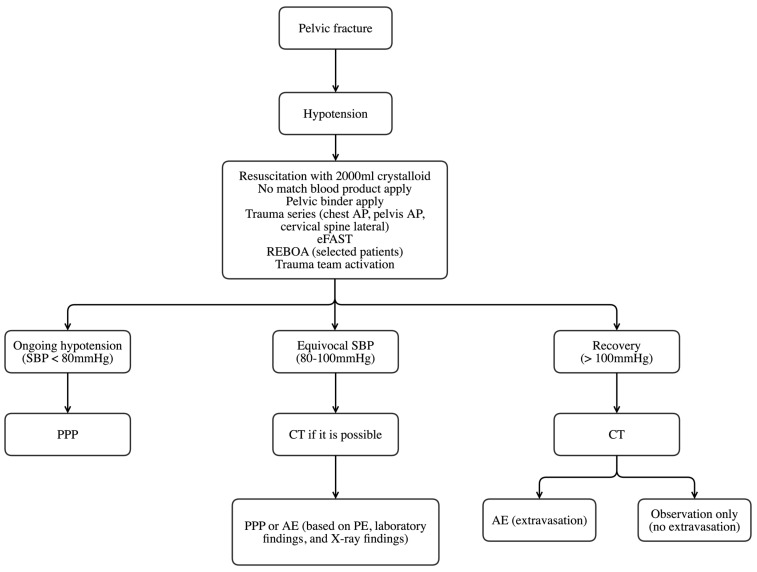
Treatment algorithm for patients with hemodynamically unstable pelvic fractures. AP, anteroposterior view; eFAST, extended focused assessment with sonography for trauma; REBOA, resuscitative endovascular balloon occlusion of the aorta; SBP, systolic blood pressure; PPP, preperitoneal pelvic packing; CT, computed tomography; AE, angioembolization; PE, physical examination.

**Figure 3 healthcare-11-01784-f003:**
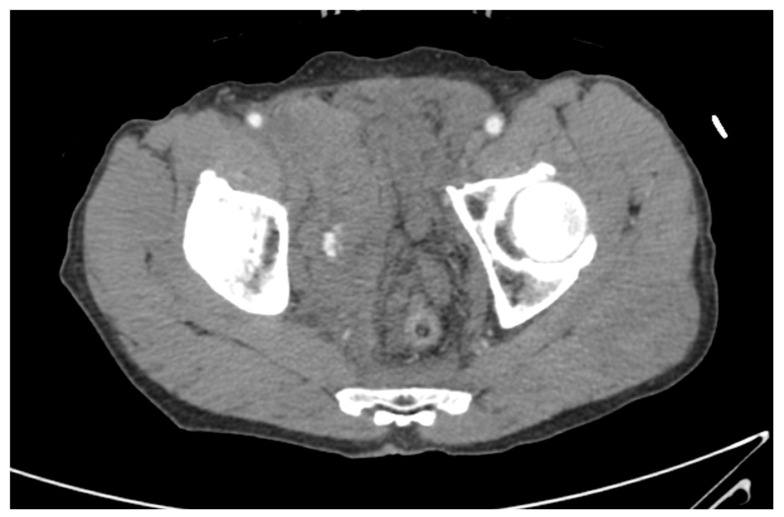
Computed tomography for a patient with a hemodynamically unstable pelvic fracture. Computed tomography showed contrast extravasation in the pelvic cavity.

**Figure 4 healthcare-11-01784-f004:**
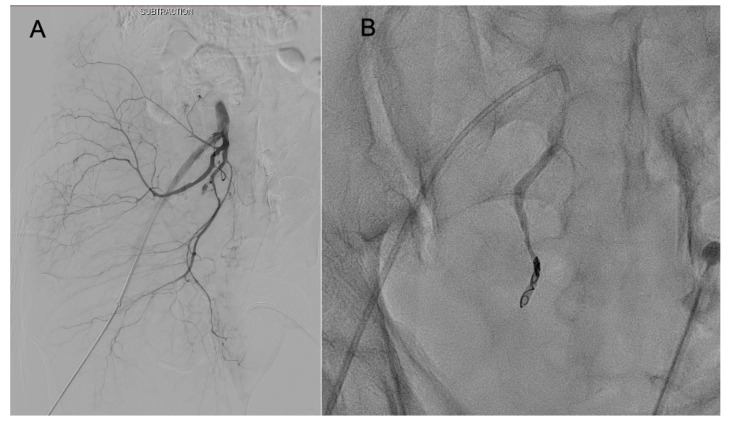
Angiography for a patient with a hemodynamically unstable pelvic fracture. (**A**) Angiography showed contrast extravasation from the pudendal branch of the internal iliac artery, and (**B**) the pudendal branch was selected and embolized with a coil.

**Table 1 healthcare-11-01784-t001:** Baseline characteristics of the enrolled patients.

	AE (*n* = 22) (%)	PPP (*n* = 43) (%)	*p*-Value
Age	58.3 ± 18.2	59.6 ± 18.3	0.782
Sex (male)	13 (59.1)	27 (62.8)	0.983
Initial SBP	78.8 ± 30.3	75.3 ± 32.6	0.668
Worst SBP in ED	66.3 ± 22.2	63.8 ± 20.9	0.663
SBP after fluid resuscitation	90.8 ± 7.7	92.1 ± 7.0	0.512
PR (median [IQR])	105.5 [91.2, 121.5]	110.0 [97.5, 126.0]	0.637 ^a^
DM	4 (18.2)	7 (16.3)	1.000 ^b^
Anticoagulant intake	3 (13.6)	4 (10.3)	0.695 ^b^
Mechanism			0.252 ^b^
Traffic accident	12 (54.5)	14 (32.6)	
Pedestrian accident	4 (18.2)	14 (32.6)	
Fall	6 (27.3)	12 (27.9)	
Other blunt trauma	0 (0.0)	3 (7.0)	
Associated injury (AIS ≥ 3)	18 (81.8)	36 (83.7)	1.000 ^b^
Abbreviated injury scale			
Head and neck	3 (13.6)	10 (23.3)	0.516 ^b^
Face	0 (0.0)	2 (4.7)	0.545 ^b^
Chest	12 (54.5)	19 (44.2)	0.597
Abdomen	8 (36.4)	13 (30.2)	0.826
Extremity and pelvic girdle	21 (95.5)	43 (100.0)	0.338 ^b^
External	0 (0)	0 (0)	NA
ISS (median [IQR])	34.0 [27.5–38.0]	38.0 [29.0–44.0]	0.188 ^a^
Initial hemoglobin level	11.7 ± 2.4	10.7 ± 2.5	0.108
Initial lactate level (median [IQR])	4.1 [2.8–6.4]	5.0 [3.9–8.4]	0.188 ^a^
Young–Burgess classification			0.282 ^b^
LC1	1 (4.5)	0 (0)	
LC2	8 (36.4)	13 (30.2)	
LC3	8 (36.4)	11 (25.6)	
APC1	1 (4.5)	0 (0)	
APC2	1 (4.5)	5 (11.6)	
APC3	1 (4.5)	3 (7.0)	
VS	2 (9.1)	11 (25.6)	
Tile classification			0.281
B	17 (77.3)	26 (60.5)	
C	5 (22.7)	17 (39.5)	
Open fracture	1 (4.5)	3 (7.0)	1.000 ^b^
REBOA	3 (13.6)	2 (4.7)	0.326 ^b^
Hybrid room	1 (4.5)	6 (14.0)	0.408 ^b^
Explo-laparotomy	3 (13.6)	8 (18.6)	0.737 ^b^
EF	1 (4.5)	9 (20.9)	0.145 ^b^
ORIF	10 (45.5)	19 (44.2)	1.000

^a^ result according to Mann–Whitney test. ^b^ result according to Fisher’s exact test. AE, angioembolization; PPP, preperitoneal pelvic packing; SBP, systolic blood pressure; ED, emergency department; PR, pulse rate; IQR, interquartile range; DM, diabetes mellitus; AIS, Abbreviated Injury Scale; NA, not applicable; ISS, injury severity score; LC, lateral compression; APC, anterior–posterior compression; VS, vertical shear; REBOA, resuscitative endovascular balloon occlusion of the aorta; EF, external fixation; ORIF, open reduction and internal fixation.

**Table 2 healthcare-11-01784-t002:** Outcomes of the enrolled patients.

	AE (*n* = 22) (Median [IQR])	PPP (*n* = 43) (Median [IQR])	*p*-Value
Time from injury to ED (minutes)	94.0 [65.8–137.2]	146 [74.5–222.0]	0.197 ^a^
Time from ED to ICU (minutes)	194.0 [146.2–269.0]	157.0 [103.5–225.0]	0.083 ^a^
Time from ED to procedure (minutes)	139.0 [95.0–155.2]	63.0 [47.0–95.0]	<0.001 ^a^
Time from ED to ORIF (days)	8.4 ± 5.9	8.1 ± 5.0	0.876
Duration of ED stay (minutes)	124.0 [84.2–221.5]	59.0 [43.5–96.0]	<0.001 ^a^
ICU LOS (days)	6.0 [2.2–12.8]	9.0 [4.0–16.5]	0.230 ^a^
Hospital LOS (days)	29.5 [4.8–49.8]	26.0 [8.5–63.0]	0.901 ^a^
PRBC transfusion requirement within 24 h (units)	8.5 [4.0–12.8]	9.0 [4.0–12.5]	0.818 ^a^
Duration of mechanical ventilation (days)	2.0 [0.2–8.0]	6.0 [2.5–14.0]	0.046 ^a^
Complications	10 (45.5%)	20 (46.5%)	1.000
Infectious complications	4 (18.2%)	15 (34.9%)	0.266
Mortality	7 (31.8%)	16 (37.2%)	0.876
Mortality due to hemorrhage	5 (22.7%)	6 (14.0%)	0.487 ^b^

^a^ result according to Mann–Whitney test. ^b^ result according to Fisher’s exact test. AE, angioembolization; IQR, interquartile range; PPP, preperitoneal pelvic packing; ED, emergency department; ICU, intensive care unit; ORIF, open reduction and internal fixation; LOS, length of stay; PRBC, packed red blood cell.

**Table 3 healthcare-11-01784-t003:** Details of patients who underwent AE after REBOA insertion.

Age	Sex	Mechanism	SBP (Initial/Worst) (mmHg)	SBP Recovered After REBOA (mmHg)	PR (/min)	ISS	YB Classification	Time from REBOA Insertion to Removal	Time from REBOA Inflation to Deflation	Location of REBOA Ballooning	AE Details	Embolized Artery	Transfusion Amount ≤ 24 h (Packed RBC)	ICU Stay	HS	Complication	Mortality
57	M	TA	80/68	95	102	38	LC2	220	51	Zone III	Unilateral selective	IIA branch	8	10	52	Pneumonia	No
70	M	TA	64/54	88	88	25	APC3	740	38	Zone III	Bilateral selective	IIA branch	9	15	23	Atelectasis	No
74	F	Pedestrian accident	81/73	100	88	22	LC1	57	14	Zone III	Bilateral selective	IIA branch	2	3	10	No	No

AE, angioembolization; REBOA, resuscitative endovascular balloon occlusion of the aorta; SBP, systolic blood pressure; PR, pulse rate; ISS, injury severity score; YB, Young–Burgess; RBC, red blood cell; ICU, intensive care unit; HS, hospital stay; TA, traffic accident.

## Data Availability

The datasets used and/or analyzed during the current study are available from the corresponding author upon reasonable request.

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
