# Peer review of "Preperitoneal Pelvic Packing versus Angioembolization for Patients with Hemodynamically Unstable Pelvic Fractures with Pelvic Bleeding: A Single-Centered Retrospective Study"

_healthcare, 2023, doi:10.3390/healthcare11121784_

Round 1
Reviewer 1 Report
Dear Authors.
Icongratulte you on your manuscript. It is a very simple analysis of your treatment protocol for treatment of hemodynamicaly unstable patients with pelvis fractures.
Anyhow the tratment of these patients has changed in the last years with more precise pharmacological treatment of bleeding in cases of blunt trauma that result in pelvic fracture. The emphasis is on resuscitation with approprate medications that stop the bleeding and not just by adding 2000 ml crystaloid solution and wait for the patient to become hemodynamicaly stable.
REBOA - during the manuscript you are writing a lot about reboa but it is not included in your sheme of action. REBOA is ment to stop the bleeding until AE can be done, which means it is basicaly used insted of PPP? Am I correct.
The place of REBOA should be placed in your basic sheme - when to use it.
Please addrese these questions in your manuscript
Author Response
Thank you very much for your insightful comment.
We provide a detailed response to each of your points.
Please see the attachment.

Reviewer 2 Report
Preperitoneal Pelvic Packing Versus Angioembolization for Patients with Hemodynamically Unstable Pelvic Fractures with Pelvic Bleeding: A Single-Centered Retrospective Study
The authors retrospectively studied the outcomes of 65 haemodynamically unstable patients (40 males, with a mean age of 59.2 years) with pelvic fractures (with a systolic blood pressure of 80–100 mmHg after initial fluid resuscitation), treated at their single centre between April 2014 and December 2022. 43 patients underwent preperitoneal pelvic packing (PPP) and 22 underwent angioembolization (AE). 3 patients (13.6%) underwent successful AE after resuscitative endovascular balloon occlusion of the aorta (REBOA) placed in zone III. The patients’ characteristics and outcomes were collected.
The authors found that the median time from emergency department (ED) to procedure and the median duration of ED stay were significantly longer in the AE group than in the PPP group (p ≤ 0.001 for both).
The median mechanical ventilation (MV) duration was significantly shorter (p = 0.046) in the AE group.
The number of patients with complications, mortality, and mortality due to haemorrhage was no different between the two groups.
The authors concluded that AE might be beneficial for patients with hemodynamically unstable pelvic fractures who show equivocal vital signs after initial fluid resuscitation, as the duration of mechanical ventilation was less and the incidence of infectious complications was lower.
This was a nice descriptive paper of a cohort of haemodynamically unstable patients with pelvic fractures. Much of the analysis and comparison between the PPP and AE groups is unfortunately of limited value as this was a retrospective and non-randomised study. Nevertheless the study has at least demonstrated that both modalities are broadly speaking equally helpful.
I have a few questions.
1)
In their series, what determined which patients had PPP and which had AE, given that this was one single Centre? Was this solely based on their BP? Were other factors at play?
In Line 130 the authors state, “AE or PPP was determined based on physical examination, laboratory data, and 130 X-ray findings, including CT findings”. This is rather vague. Please can the authors elaborate.
2)
Did any patients have PPP and then go to later have AE? This does happen as not all pelvic bleeding is packable.
3)
What determined which patients had REBOA?
4)
Can the Authors add where REBOA fitted into their treatment algorithm in Figure 2
5)
Given that patients were not randomised to PPP and AE, and that the criteria for choosing PPP or AE were based on how shocked the patients were, it does not seem so logical to directly compare the outcomes of the patients. These 2 groups of patients were in different haemadynamic states.
Can the authors comment?
6)
There needs to a full edit and spell check, for example in Figure 1
“who did not underwent”, should be “who did not undergo”.
There needs to a full edit and spell check, for example in Figure 1
“who did not underwent”, should be “who did not undergo”.
Author Response

(The authors gave the same response as above.)
